# Aerial Drone Surveys Reveal the Efficacy of a Protected Area Network for Marine Megafauna and the Value of Sea Turtles as Umbrella Species

Liam C. D. Dickson [1], Stuart R. B. Negus [1], Christophe Eizaguirre [1], Kostas A. Katselidis [2] and Gail Schofield [1,*]

1   School of Biological and Behavioural Sciences, Queen Mary University of London, Mile End Road, London E1 4NS, UK
2   National Marine Park of Zakynthos, 1 El. Venizelou Str., 29100 Zakynthos, Greece
*   Correspondence: gail.schofield@qmul.ac.uk

**Abstract:** Quantifying the capacity of protected area networks to shield multiple marine megafauna with diverse life histories is complicated, as many species are wide-ranging, requiring varied monitoring approaches. Yet, such information is needed to identify and assess the potential use of umbrella species and to plan how best to enhance conservation strategies. Here, we evaluated the effectiveness of part of the European Natura 2000 protected area network (western Greece) for marine megafauna and whether loggerhead sea turtles are viable umbrella species in this coastal region. We systematically surveyed inside and outside coastal marine protected areas (MPAs) at a regional scale using aerial drones (18,505 animal records) and combined them with distribution data from published datasets (tracking, sightings, strandings) of sea turtles, elasmobranchs, cetaceans and pinnipeds. MPAs covered 56% of the surveyed coastline (~1500 km). There was just a 22% overlap in the distributions of the four groups from aerial drone and other datasets, demonstrating the value of combining different approaches to improve records of coastal area use for effective management. All four taxonomic groups were more likely to be detected inside coastal MPAs than outside, confirming sufficient habitat diversity despite varied life history traits. Coastal habitats frequented by loggerhead turtles during breeding/non-breeding periods combined overlapped with 76% of areas used by the other three groups, supporting their potential use as an umbrella species. In conclusion, this study showed that aerial drones can be readily combined with other monitoring approaches in coastal areas to enhance the management of marine megafauna in protected area networks and to identify the efficacy of umbrella species.

**Keywords:** functional diversity; indicator species; ocean health; spatial prioritisation; UAS; UAV

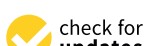



## 1. Introduction

When designing protected areas (PAs), "hotspots" supporting high species richness and/or abundance of threatened wildlife are generally targeted [1,2]. Such areas usually aim to encompass the habitat needs of key life history stages for target species, particularly when distinct resources are required (e.g., reef fishes with pelagic larvae [3], coastal and oceanic foraging in breeding and non-breeding albatrosses [4]). The most effective PAs tend to be large, isolated, well established and properly enforced [5,6]. However, with human pressure on space and resources steadily increasing, meeting these criteria is challenging [7,8]. At present, 7.9% of global marine environments are covered by marine protected areas (MPAs), with the percentage being slightly higher close to shore (18.5% of coastal areas) [9]. The challenge of protecting global ocean environments has led to the development of networks of multiple smaller MPAs, which expand total coverage and increase connectivity and coverage of multiple taxa [8,10,11]. Implementing PAs that capture the needs of multiple taxa, rather than single targeted species, is important to

maintain healthy ecosystems [12–14], leading to many studies evaluating the effectiveness of species currently used as umbrellas for other wildlife (i.e., when one species and its habitats serves as a surrogate for the protection of other co-occurring species), towards developing robust approaches for systematic conservation planning [15,16].

Ideally, PAs should encompass the year-round distributions of focal species; however, this is difficult for marine megafauna for which migrations and life stages often cover entire ocean basins [12,17]. While some species aggregate in large numbers at breeding or foraging grounds in predictable locations (e.g., pinnipeds [18], sea turtles [19]), other species are mostly pelagic and are often solitary (e.g., whales [20], sharks [21]). Consequently, researchers use various approaches to monitor abundance and distribution, including direct observations (e.g., sighting, stranding, capture–mark–recapture data) and remote monitoring (e.g., loggers or satellite and acoustic telemetry). This has led to the accumulation of large datasets on these groups [22–24]. Despite the invaluable insights garnered through such studies, all of these approaches have inherent biases in location, scale or sample size, which generate uncertainty in representativeness of the data [25,26], making it difficult to assimilate data across multiple groups [8,12]. Unoccupied Aircraft Systems (UAS; termed aerial drones hereafter) could help bridge this gap by the capacity to survey multiple species at once at various scales, complementing existing monitoring approaches [14,27–29]. By monitoring areas both inside and outside protected areas objectively, the optimal habitat of both target species and non-target species could be detected [2,8], facilitating more comprehensive and holistic conservation initiatives.

Marine megafauna are considered indicators of ocean health due to their vulnerability to human activity [30], their association with multiple distinct habitats (e.g., breeding and foraging grounds) and their potential to indicate underlying prey distributions and ecosystem processes [15,31]. Western Greece is a biodiversity hotspot, supporting some of the highest nesting densities in the Mediterranean for loggerhead sea turtles (*Caretta caretta*), but also supporting the second highest number of threatened species in Europe [32,33]. Examples include pinnipeds (critically endangered Mediterranean monk seal, *Monachus monachus*), cetaceans (e.g., fin whales *Balaenoptera physalus*, Cuvier's beaked whale *Ziphius cavirostris*) and elasmobranchs (e.g., blue sharks *Prionace glauca*, basking sharks *Cetorhinus maximus*, bull rays *Aetomylaeus bovinus*). The Natura 2000 network of protected areas was established in 1992 to protect the habitats of threatened wildlife in line with the IUCN [33], including in the marine environment, and is now the largest coordinated network globally covering >8% of marine territory in the European Union (EU) [34]. Sites are established with the goal of conserving specific species and habitats listed in the EU Habitats Directive and Birds Directive, based in part on conservation status (e.g., endangered, vulnerable [33]) and cultural importance. However, many sites were originally established using a precautionary approach during the 1990s and early 2000s when holistic region-scale monitoring approaches were restricted to costly manned plane surveys, remote tracking was still in its infancy and commercial aerial drones had yet to emerge [35,36]. Thus, systematic assessments are still required to evaluate whether current protection effort captures different life history requirements of key wildlife, including marine megafauna, as stipulated in the recent United Nations Program IMAP RAC-SPA Quality Status Report on ocean health in the Mediterranean basin [37].

Here, we evaluated whether part of the European Natura 2000 protected area network (coastal region of western Greece) encompasses the habitat needs of four marine megafauna taxa (sea turtles, elasmobranchs, cetaceans and pinnipeds). We used systematic region-wide aerial drone surveys to obtain baseline data on marine megafauna presence inside and outside coastal MPAs, then integrated published datasets (remote tracking, sightings, strandings) on the four taxa. We expected that aerial drone surveys would provide complementary information on the distributions of marine megafauna across the network (i.e., inside vs. outside PAs) to existing monitoring approaches. We hypothesized that the coastal network in our study region would better support the breeding habitat requirements of loggerhead sea turtles compared to the non-breeding habitat requirements

(i.e., developmental, foraging and wintering habitats) or areas outside of the network. We also hypothesized that the coastal network in our study region would better protect sea turtles over other marine megafauna, with loggerhead sea turtles serving as viable umbrella species. We expect our results to demonstrate the value of region-wide aerial drone surveys as a platform for integrating data from diverse approaches on multiple marine megafauna and evaluating wildlife distributions with greater certainty to inform current and future conservation efforts in coastal areas.

## 2. Materials and Methods

### 2.1. Study Region and Species

This study was conducted in the Ionian region of western Greece, including Zakynthos Island, Kefalonia Island and the western and southern coasts of the Peloponnese (Figure 1). The selected study region (1508 km coastline) encompassed 12 coastal Natura 2000 designated MPAs [38] (56% of surveyed coastline), of which two are designated national parks (National Marine Park of Zakynthos and Kotychi-Strofylia Wetlands National Park). The Natura 2000 network aims to conserve the habitats used by threatened plant and animal species. This region supports important habitats for sympatric marine megafauna, including threatened sea turtles, elasmobranchs, cetaceans and pinnipeds. The region supports three major loggerhead sea turtle rookeries (>100 nests/season; Laganas Bay: 1248 nests/season; Kyparissiakos Bay: 881 nests/season; Lakonikos Bay: 197 nests/season) [39], as well as minor to intermediate rookeries, which are fully or partly encompassed by the Natura 2000 network, and supports in-water turtle densities of up to 270 turtles/km during the breeding period (April to August) [40]. Both green and loggerhead sea turtles have been documented foraging year-round in this region, including both adult and immature turtles on Zakynthos, Kefalonia and across the Peloponnese [41,42]. Greek waters support 67 species of elasmobranchs, including 42 documented shark species, which face decline and local extinction throughout the region as a result of overfishing [43]. Twelve species of whales and dolphins are also present throughout the region, with seven being permanently present and commonly observed (striped dolphin, common bottlenose dolphin, short-beaked common dolphin, Cuvier's beaked whale, sperm whale, Risso's dolphin and fin whale) [44]. Pinnipeds include the critically endangered Mediterranean monk seal, with small numbers of individuals primarily inhabiting areas of Zakynthos and Kefalonia [45].

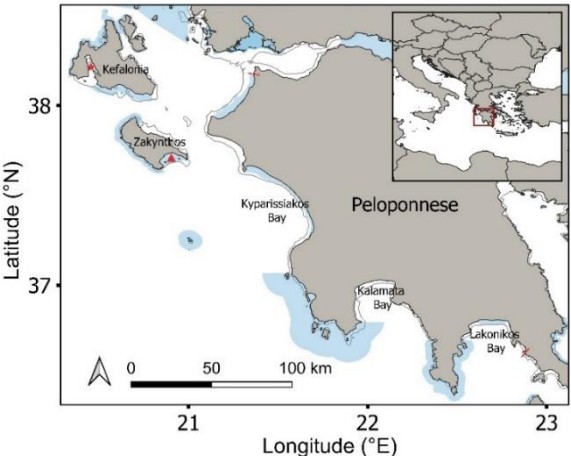

**Figure 1.** Study region including the Natura 2000 protected area network (shaded blue polygons). Inset map shows study area within the wider Mediterranean. Red star: Gulf of Argostoli; Red triangle: Laganas Bay; Red lines show coastal limits of aerial drone surveys; Kato Korogona in the south to Kalogria in the north. Surveys spanned 0–400 m offshore.

## 2.2. Aerial Drone Surveys and Data Processing

From 2016 to 2020, aerial drone surveys were conducted in the coastal areas spanning ~620 km of the coastline of the western and Southern Peloponnese (Kato Korogona in the south to Kalogria in the north, indicated by red lines on Figure 1), Zakynthos Island (Laganas Bay) and the southern part of Kefalonia (including Argostoli Gulf and south coast). In 2019 and 2020, aerial drone surveys were conducted systematically (homogenous survey effort) across these locations over one month during the breeding (May–June) and non-breeding (September–October) periods of sea turtles [46]. The 2016–2018 surveys were conducted intermittently over different months and were used to confirm consistency in distributions at the same sites over multiple years. The aerial drone (DJI Phantom 3 Professional™; Shenzhen, China; http://www.dji.com, accessed on 10 January 2022) was flown at 60 m altitude (giving 100 m field of view), with transects set at 100 m intervals up to 400 m offshore (for details, see [40]). Data on aerial drone surveys for the 2016–2019 breeding period were previously published in Dickson et al. [40], while the non-breeding period data have not been published. The geographical position (longitude and latitude) of all megafauna (sea turtles, elasmobranchs, cetaceans and pinnipeds) detected in drone footage was recorded manually by at least two independent observers [47,48]. The coastline of the study region was divided into 2 km × 400 m sections (termed cells hereafter), to allow sufficiently fine-scale evaluation of the effect of extending MPA coverage. QGIS (QChainage plugin) was used to delineate each 2 km section, with 754 cells of the 1508 km coastline being automatically generated by the program. For each cell, we recorded the number of animals present, protection status (protected/unprotected) and habitat type. We validated the detection of animals with aerial drone surveys in a previous study by ground truthing with parallel boat surveys [49]. The presence of individual animals, species (where possible) and group (four taxa) on drone footage was validated by at least two trained observers independently. Animals from the four marine megafauna groups were distinguished based on their distinct body forms [48]. Four general habitat types were delineated: vegetated, submerged sandbank, rocky outcrops and reefs and mudbank/silt. Habitat type was delineated using aerial drone surveys and Google Earth images, which were ground truthed at drone surveyed sites with boat and/or swim/snorkel surveys. No discrepancy was detected across approaches for the four groupings.

## 2.3. Processing of Published Datasets

We assembled a database on the distributions of sea turtles, elasmobranchs, cetaceans and pinnipeds in the study area from research articles published in peer-reviewed scientific journals and open-access Internet databases (Global Biodiversity Information Facility, Ocean Biodiversity Information System) (Table S1 [44,50–66]). For remotely tracked animals, we recorded information provided by various sources (including species, life history stage, adult/juvenile), sex, period of tracking) on cells on Google Earth and QGIS [67]. For sea turtles, animals were tracked from various locations across the Mediterranean. For the purposes of our study, we excluded migratory movement of sea turtles, as we focused on identifying habitats used during breeding and non-breeding periods only. For turtles not tracked with Fastloc GPS, we only used the central three cells of each home range to correspond to the mean 50% KUD of those recorded from Fastloc GPS tracked turtles (accurate to within 20 m) (to limit overestimates from less accurate devices) [68]. Due to the scarcity of tracking data in the study region for the other three taxa in our region, we included visual observations and strandings in coastal areas (excluding ports) as a proxy. Total numbers of species and total numbers of animal groups for each coastal section were assimilated.

## 2.4. Data Analysis

We assessed how published tracking datasets compared to aerial drone surveys in capturing the distributions of loggerhead sea turtles along the coastline of the study region. We used Chi-square analysis to evaluate the number of cells with turtle presence/absence during breeding and non-breeding periods separately and combined for the two data

sources. We also assessed how densities of turtles within cells varied across the two data sources. To remove bias due to false negatives, cells that were not surveyed were excluded from the density analysis. We then integrated the two data sources to obtain the following data for each cell: (1) overall presence/absence (2 levels: present, absent); and (2) seasonal presence/absence (4 levels: absent breeding and non-breeding, present breeding only, present non-breeding only, present both breeding and non-breeding period). We examined: (1) MPA coverage of breeding turtles, non-breeding turtles and both groups combined; and (2) habitat types used by the three groups inside and outside MPAs. Non-breeding turtles were considered to be occupying developmental, foraging, or wintering habitat.

For all four marine megafauna taxa, presence/absence data (to remove bias toward sea turtle records) were used to characterise cells based on the number of taxa present (i.e., 0–4, with respect to sea turtles, elasmobranchs, cetaceans and pinnipeds). This was repeated for individual species to check if generated results differed (6 groupings were used as 6 species maximum were recorded in a single cell). We examined: (1) coastal MPA coverage of the four taxa based on the total number of taxa present in each cell; and (2) whether the habitat types used inside and outside coastal MPAs could be attributed to individual taxa or total taxa in each cell. We also recorded the cells in which no animals were recorded in the (1) integrated dataset; and (2) aerial drone surveys and other sources. Analyses were repeated with and without these cells to detect any effects on outputs.

Finally, we examined how the spatial coverage of the coastal MPA network could be enhanced with minimal changes. To do this, we evaluated the number of cells outside coastal MPAs that needed to be added to attain 100% protection of all 4, 3, 2 and 1 taxon combined. This was repeated for species to detect any differences. We also quantified the effectiveness of using loggerhead sea turtles as an umbrella species for multiple taxa in the breeding and non-breeding seasons, both inside and outside of coastal MPAs (i.e., the percentage of other taxa that were in areas containing sea turtles).

### 2.5. Statistical Analyses

We performed all statistical analyses in RStudio [69] and created all density distribution and habitat maps in QGIS [67]. We used Chi-square analysis to investigate the relationship between the protection status of sites and habitat type for each of the four habitat types, protection status and each taxon (cetaceans, sea turtles, elasmobranchs and pinnipeds) and percentage protection provided by sea turtles for other marine megafauna under different scenarios. Chi-square goodness of fit tests were used to analyse the primary habitat of each taxon based on cell counts. We also applied Chi-square analysis to test the relationship of turtle density to habitat type and protection status. Where Chi-square approximations returned estimation warnings, we used Fisher's Exact Test.

## 3. Results

### 3.1. Assimilated Data

Overall, we obtained 18,059, 209, 25 and 2 records of sea turtles, elasmobranchs, cetaceans and pinnipeds in the aerial drone surveys in 152 cells out of 168 surveyed cells (from the 754 generated 2 km coastal cells) in the coastal study region. Records on the four taxa from other sources were obtained in 378 cells of the coastal region (Table S1; individual groups were recorded in 212, 22, 217 and 4 cells, respectively). When data from aerial drones and other survey types were combined, animals from the four taxa were present in 435 cells, with just 22% overlap for drone surveys and other survey types. Of the 319 cells containing no animals, 16 were surveyed by aerial drones documenting absence in these cells only.

### 3.2. Habitat Types Inside and Outside Coastal MPAs

The most common coastal habitat type across the region is reefs and rocky outcrops (56%), followed by submerged sandbanks (28%) and vegetated areas (14%) (Figure 2a,b). Coastal MPAs cover approximately 50% of each habitat type (vegetated: 47%; submerged

sandbank: 59%, reefs and rocky outcrops: 57%; mud/silt: 53%) (Figure 2b), with no statistical significance between habitat type and current protection ($\chi^2$ = 4.22, df = 3, *p* = 0.2).

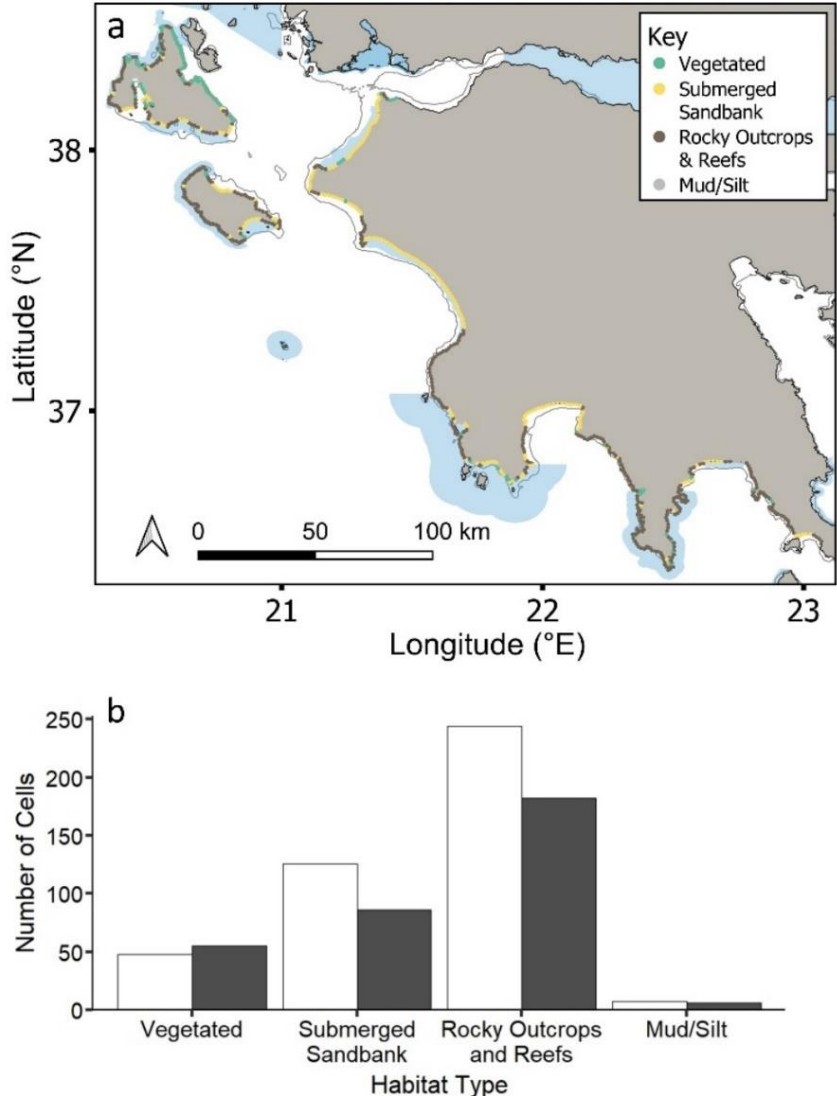

**Figure 2.** (**a**) Study region showing the distribution of the four key habitat types inside and outside MPAs. Natura 2000 MPAs: shaded blue polygons. Circles represent 2 km cells along coastline. (**b**) Total number of cells represented by each habitat type inside (grey) and outside (black) MPAs.

### 3.3. Sea Turtle Coverage based on Tracking Versus Aerial Drone Surveys

Loggerhead sea turtles were recorded in 30% of the 754 generated cells across the coastal study region when combining both tracking and aerial drone datasets (Figure 3a). Out of all cells containing loggerhead sea turtles, individuals were recorded in 61% and 67% of cells for tracking datasets (156 individuals spanning 16 years) and aerial drone surveys, respectively (presence/absence: Figure 3a; Figures S1 and S2); however, there was just a 29% overlap, with sea turtles being in nine more cells for aerial drone surveys (Figure 3a,b). This overlap was similar (35%) when evaluating breeding and non-breeding cells separately (Figure 3b).

The maximum number of loggerhead turtles recorded in a given cell was 164 and 58 for drone and tracked turtles, respectively. This difference was expected due to the upper limit in the number of tracked individuals (*n* = 156 turtles). Cells that overlapped for the two approaches were more likely to be those where a high number of loggerhead sea turtles had been tracked (Figure 3c; tracking, $\chi^2$ = 20.6, df = 3, *p* < 0.001; aerial drone

surveys, $\chi^2 = 8.3$, df = 3, $p < 0.05$). Turtle density in the same cells corresponded for the two approaches up to 100 turtles, declining with increasing density (Figure 3c; Figure S3).

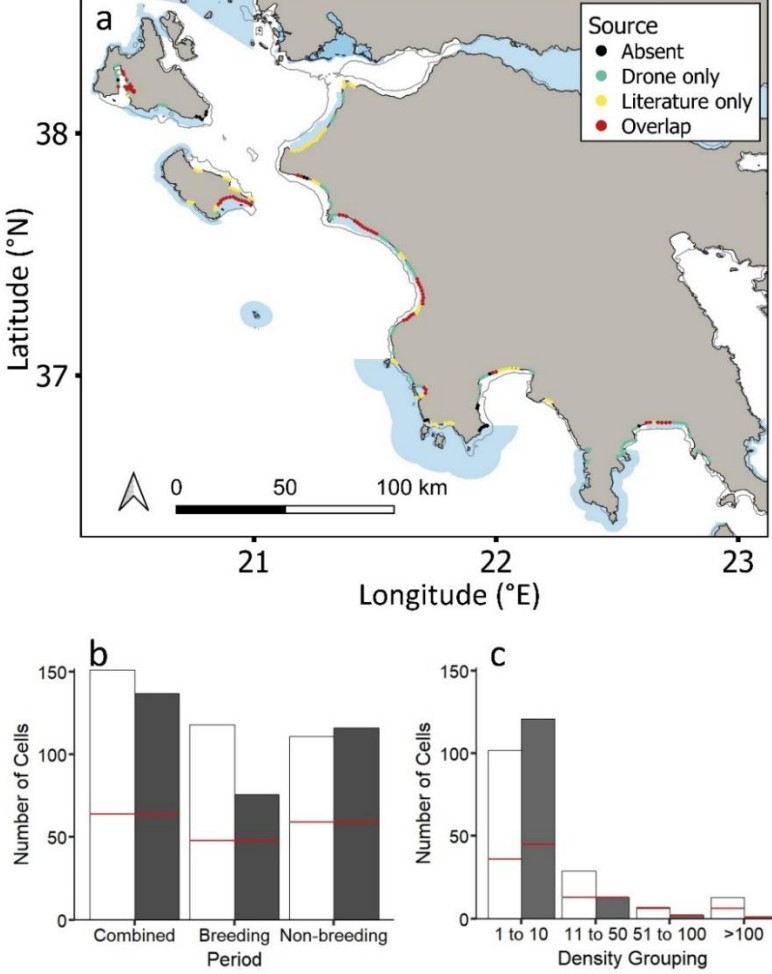

**Figure 3.** (**a**) Study region showing the loggerhead sea turtles (*Caretta caretta*) recorded in aerial drone surveys and tracking datasets and both aerial and tracking datasets (for each approach separately see Figure S1). Natura 2000 MPAs: shaded blue polygons. Circles represent 2 km cells along the coastline. (**b**) Loggerhead sea turtles recorded in cells for aerial drone surveys (white bars) versus tracking datasets (grey bars) during the combined breeding/non-breeding period, the breeding period and the non-breeding period. (**c**) Number of cells containing loggerhead sea turtles of different densities for aerial drone surveys (white bars) and tracking datasets (grey bars) (Table S1). Red lines in (**b**) and (**c**) show the number of cells for which tracking and aerial drone surveys overlap.

### 3.4. Sea Turtle Distributions inside and outside MPAs

Over 99.6% of all sea turtle records were loggerheads, with the remainder being green turtles (thus all individuals were combined in the analyses). Green turtles were widely dispersed throughout the region at low densities (*n* = 50 cells; 1–3 green turtles/cell). Loggerhead sea turtle densities were higher during the breeding period compared to the non-breeding period, with a maximum of 154 sea turtles in a cell (breeding) compared to 119 (non-breeding). However, sea turtles had similar distributions in both periods (Figure 4a). Out of all cells containing turtles, 65% and 75% of cells contained turtles during breeding and non-breeding periods, respectively, with 41% overlap (Figure S4). More cells within coastal MPAs (59%) contained overlapping breeding/non-breeding cells compared to outside coastal MPAs (Figure S4; $\chi^2 = 14.7$, df = 3, $p < 0.005$).

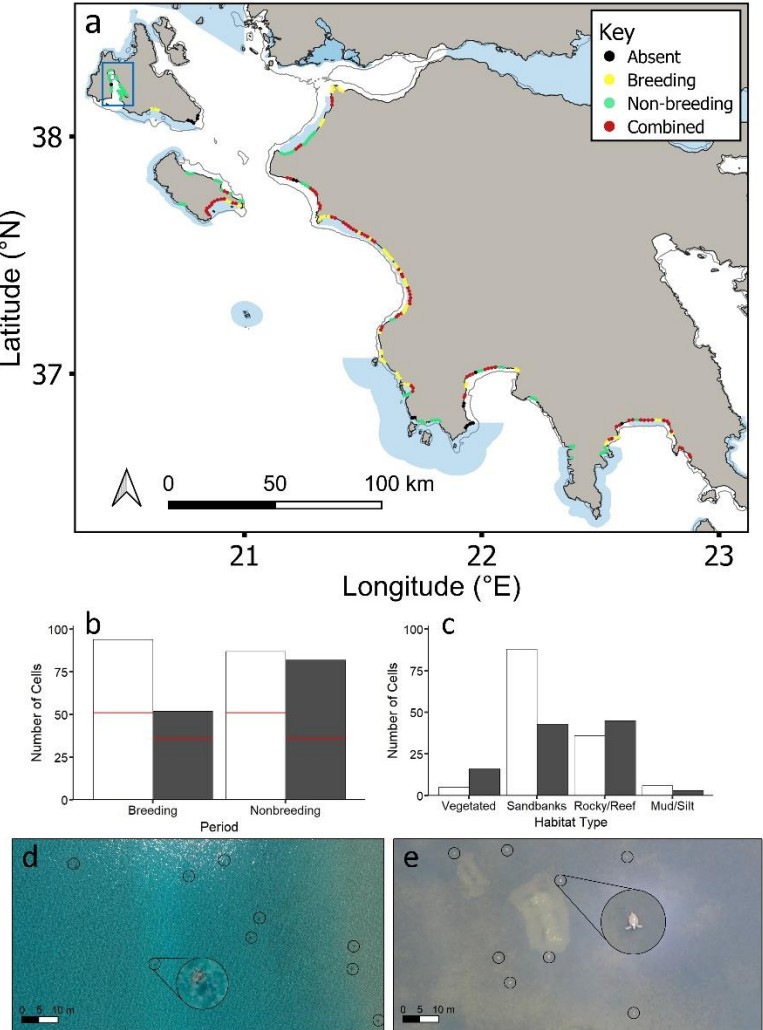

**Figure 4.** (**a**) Study region showing where loggerhead sea turtles (*Caretta caretta*) were recorded during the breeding period, non-breeding period and both periods combined. Data are integrated from aerial drone surveys and the published literature (Table S1). Figure S2 presents separate breeding and non-breeding distributions. (**b**) Loggerhead sea turtles recorded in cells during the breeding period and non-breeding period inside (white bars) and outside (grey bars) MPAs. The number of overlapping breeding/non-breeding cells is shown by a red line. (**c**) Number of cells of each habitat type over which loggerhead sea turtles were recorded in each period inside (white bars) and outside (grey bars) MPAs for overlapping breeding/non-breeding period cells (the same trends were obtained for all breeding period only cells and non-breeding period only cells; Figure S5b,e). Aerial drone images of sea turtles (**d**) over submerged sandbanks during the breeding period, and (**e**) over vegetated substrate during the non-breeding period. Sea turtles are circled in black. Enlarged circles show turtles at 5× magnification. Natura 2000 MPAs: shaded blue polygons. Circles represent 2 km cells along the coastline. Blue rectangle represents Gulf of Argostoli (Kefalonia), which supports year-round non-breeding habitat (for density and habitat type see Figures S5c,f and S6).

The distribution of loggerhead sea turtles across coastal habitats was similar during breeding and non-breeding periods (Figures S5 and S6). Inside coastal MPAs, cells containing breeding and non-breeding turtles were primarily submerged sandbank habitat (breeding: $\chi^2 = 86.1$, df = 3, $p < 0.001$; non-breeding: $\chi^2 = 46.1$, df = 3, $p < 0.001$), including overlapping breeding/non-breeding cells ($\chi^2 = 45.1$, df = 3, $p < 0.001$; Figure 4c). Outside coastal MPAs, cells containing breeding and non-breeding sea turtles were also primarily submerged sandbank habitats ($\chi^2 = 8.0$, df = 3, $p < 0.05$; $\chi^2 = 10.4$, df = 3, $p < 0.05$). Cells that were surveyed for which no turtles were detected were not significantly more likely to be

inside or outside coastal MPAs, or over any particular type of habitat. The Gulf of Argostoli (Kefalonia) supported high densities of non-breeding loggerhead sea turtles during both periods (i.e., this region supported non-breeding sea turtles during the breeding period; Figure 4a). All of these cells fell outside of coastal MPAs, and the habitat type was primarily vegetated substrate (Fisher's exact test; $p < 0.05$).

### 3.5. Marine Megafauna Distributions Inside and Outside Coastal MPAs

While individuals of all four taxa were recorded throughout the study region, up to three taxa were only ever recorded in any one cell (Figure 5a). A similar pattern was recorded at the species level (Figure S7). Cells containing any of the four taxa (i.e., 1–3 taxa in any given cell) were more likely to be inside coastal MPAs compared to outside (64% cells were inside; Figure 5 andFigure 6a; $\chi^2 = 10$, df = 3, $p < 0.05$), with the same result being obtained when considering each taxon separately (Figure 6b; $\chi^2 = 27.8$, df = 3, $p < 0.001$). Cells containing three or two taxa were statistically more likely to be over submerged sandbanks (three taxa: $\chi^2 = 34.1$, df = 3, $p < 0.001$; two taxa: $\chi^2 = 25.5$, df = 3, $p < 0.001$; Figure 6c). In contrast, cells containing just one taxon were not significantly linked to any one coastal habitat type (Figure 6c). Certain taxa were more significantly associated with certain coastal habitats. For instance, cetaceans and pinnipeds were more strongly associated with rocky outcrops and reefs, whereas loggerhead sea turtles and elasmobranchs were more strongly associated with submerged sandbanks (Figure 6d; all $p < 0.001$; Table S2; similar trends were obtained at the species level; Figure S7). Cells that were surveyed but contained no animals ($n = 16$ cells) mostly occurred over submerged sandbanks inside and outside of protected areas ($\chi^2 = 10.2$, df = 3, $p < 0.05$; Figure 6a,c). Figures S8 and S9 show the distributions of each taxon.

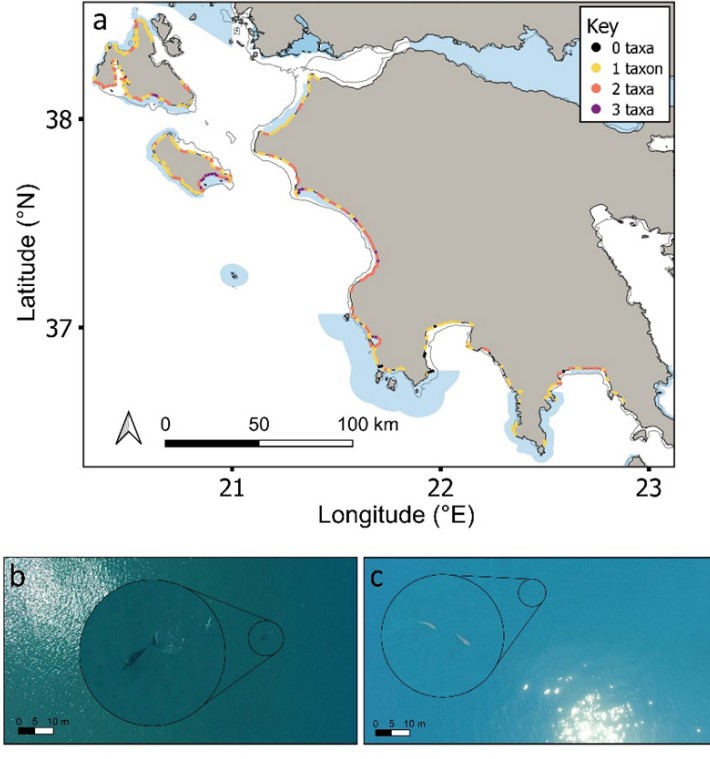

**Figure 5.** (**a**) Number of taxa found in each cell across the study region (Figures S6 and S7 for each taxon separately). Natura 2000 MPAs: shaded blue polygons. Aerial drone images of (**b**) Cuvier's beaked whale (*Ziphius cavirostris*) sighted at 100 m from shore; (**c**) sharks sighted at 400 m from shore. Enlarged circles show the animals at 5× magnification.

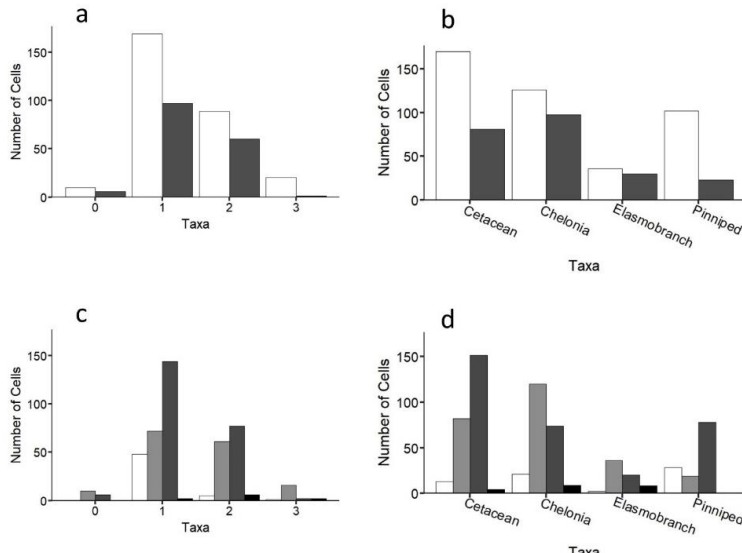

**Figure 6.** Number of taxa (**a**) inside (white bars) and outside (grey bars) MPAs and for (**b**) each taxon separately. Habitat type of cells containing (**c**) 0–3 taxa; and (**d**) each taxon separately for the entire study region. White, vegetated; light grey, submerged sandbanks; dark grey, rocky outcrops and reefs; black, mud/silt.

*3.6. Enhancing Protected Area Coverage*

The best scenario for using the loggerhead sea turtle as a possible umbrella species to protect multiple taxa was when areas frequented by loggerheads during both breeding and non-breeding periods were used (>30% better coverage, Figure 7; Figures S10 and S11), generally followed by areas frequented by loggerheads during non-breeding periods and breeding periods separately. This scenario captured 76% of multiple taxa cells (2–3 taxa within a cell; 74% and 78% inside and outside MPAs, respectively), demonstrating that the loggerhead sea turtle could be used as an umbrella species but only when combining coastal areas frequented by turtles during breeding and non-breeding periods in our study region. However, while there was no significant difference among scenarios (Table S3), caution should be taken when using turtles as umbrellas based on breeding areas alone. Protection of multiple species produced similar results to that of multiple taxa (Figures S12 and S13).

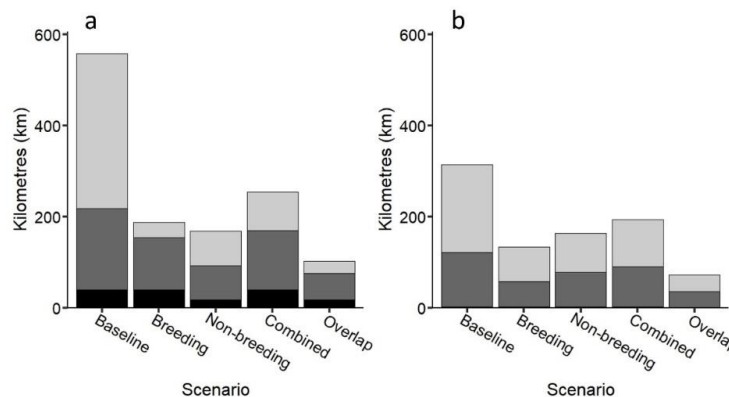

**Figure 7.** (**a**) Number of kilometres of coastline inside MPAs capturing taxa based on five scenarios: (1) all cells with one or more taxa (baseline); (2) all cells containing breeding turtles; (3) all cells containing non-breeding loggerhead turtles; (4) all cells containing combined breeding and non-breeding loggerhead turtles; (5) just overlapping cells containing breeding/non-breeding loggerhead turtles (combined). Black: 3 taxa cells; dark grey: 2 taxa cells; light grey: 1 taxon cells; (**b**) Number of additional kilometres required to extend protection effort outside MPAs based on the same five scenarios.

## 4. Discussion

This study demonstrates that the Natura 2000 PA network in the coastal region of western Greece supports multiple marine taxa, with loggerhead sea turtles as possible umbrella species. We also show that aerial drones can be readily combined with other monitoring approaches in coastal areas to enhance the management of marine megafauna to mitigate current and potential human threats.

Our large-scale drone surveys demonstrate that the network of coastal Natura 2000 PAs in our study region captured the distribution of all four marine megafauna taxa, despite their diverse life histories, movement patterns and social behaviours. This could be attributed to the network capturing >50% of the four general habitat types delineated here [5,70,71]. For instance, pinnipeds and cetaceans were more closely associated with rocky outcropping/reef type coastal habitats in our region, whereas sea turtles and elasmobranchs were more closely linked with submerged sandbank type coastal habitats, supporting existing studies [40,72]. Pinnipeds, represented by the critically endangered Mediterranean monk seal, had the most constrained distributions, with very low population sizes (around 20 individuals) in remote areas with sea caves on Kefalonia and Zakynthos [45]. In our study region, loggerhead sea turtles feed on sponges in reefs and mine for molluscs in shallow submerged sandbanks and muddy substrate, while green turtles feed on seagrass [41,42]. These sandbanks are also important for breeding turtles, which tend to aggregate over them during breeding in our study region, potentially to enhance egg development through access to thermally warmer waters [40,73].

This dual use of shallow submerged sandbanks in coastal areas for breeding and non-breeding activity (developmental, foraging and wintering) by loggerhead sea turtles was reflected in our study. Our analysis also identified areas supporting large numbers of sea turtles, where other taxa are also present, that remain unprotected, which require management/policy focus (e.g., the Gulf of Argostoli and Kalamata Bay). Various automated approaches to habitat mapping are being developed that do not require field validation, including combining remotely operated underwater vehicles (ROV) with sonar and remote sensing approaches, which could help refine our understanding of how marine megafauna use such coastal habitats [27,74,75]. Of note, while Mediterranean monk seals tend to remain resident in certain areas [45], the species of the other three taxa move more widely [71]. For instance, sea turtles may migrate more than 1000 km, while cetaceans in the Mediterranean may span distances of 200–2000 km [76] and sharks 300–3000 km [77]. Thus, even the network of coastal MPAs within our study region likely captures only part of the life history needs of these four groups, with the existing basin-wide network being key to ensure their needs are met even under climate change [5,6,14,78].

Through assimilating comprehensive baseline data on the distributions of these four taxa, consistent long-term monitoring would allow us to evaluate the effects of climate change as it accelerates [11,17,78]. Key insights on distributions, movement patterns and behaviour can be obtained from direct observations, strandings, tracking (and logging) studies, as well as aerial drones [17,22,24,28,41]. However, the current study showed that bias exists in individual monitoring approaches regarding the coastal distributions of marine megafauna, supporting existing studies calling for caution in using single approaches to infer distributions, even when assimilating data from 1000 s of animals across taxa [23,79]. For instance, even for loggerhead sea turtles, where large numbers of turtles (>100 individuals) have been remotely tracked in our study region, tracking datasets and aerial drone surveys only had 29% overlap in detected coastal area use, with this overlap being greater for non-breeding areas compared to breeding areas. This showed the complementarity of these two approaches for detecting and recording turtle presence. However, because it is possible to monitor large areas of coastline uniformly for multiple species at once and at low cost with aerial drones [36,40,75], it could prove effective for integrating various monitoring approaches and reducing the bias of any single approach. Commercial aerial drones are also biased to the areas where they can be used, with limited capacity (and greater risk) further offshore; however, they represent a potentially powerful tool to inform

conservation efforts and spatio-temporal zoning, particularly for coastal area managers and policy makers [14,27,28,75].

We showed that loggerhead turtles could work well as umbrella species for other marine megafauna in our study region, particularly when both breeding and non-breeding coastal habitats are selected for protection, and we recommend that this approach should be explored in other regions. One possible explanation is that, because sea turtle breeding areas are primarily restricted to certain coastal habitats with characteristics typically not associated with feeding, the food resources required by other taxa might not be supported as effectively [80,81]. Thus, it is important to understand what the different marine megafauna groups feed on and how these resources overlap. For instance, previous studies demonstrated the importance of what animals feed on in the ability of individual species to serve as an effective surrogate to other wildlife [80,82,83] (i.e., herbivores versus carnivores), with our study also identifying the importance of life stage. As the seven sea turtle species feed on different items (including herbivores, omnivores and carnivores), with differences also arising across life stages [80,84,85], this should also be taken into consideration when evaluating their value as umbrella species or other wildlife, including in areas where they are present during non-breeding periods [80,84,85].

The use of one species as a surrogate, or umbrella, for others faces limitations [15,16,82]. For instance, our study focused on a coastal region used by sea turtles for both breeding and non-breeding activities; however, the other taxa likely use this region for different life history needs and in relation to different parameters. Furthermore, if the distribution of the umbrella species shifts, other species must match this shift or be left unprotected [80]. Globally, all seven species of sea turtles are threatened by human activities across their entire life history, with breeding areas being targeted for protection to safeguard future generations [19,85,86]. Therefore, protecting these areas remains important, regardless of their value as conservation surrogates [12,16,80]. However, more focused effort to identify and protect non-breeding areas (including developmental, foraging and wintering habitats) is also required, which, in turn, would safeguard other key marine megafauna [12,15,24,30]. Ultimately, loggerhead sea turtles, at least in our study region, are viable umbrella species that could be used to help protect habitats used by other marine megafauna. Our study findings demonstrate the importance of the Natura 2000 network for protecting marine biodiversity and the value of region scale monitoring of marine megafauna (capturing areas inside and outside MPAs) to evaluate and propose management actions [6,12,28,79].

## 5. Conclusions

This study demonstrated the potential of integrating multiple monitoring approaches to capture the distributions of sympatric marine megafauna in coastal areas comprehensively. In particular, aerial drone surveys proved useful in linking the different approaches, as they could be used uniformly across sites, regardless of protected area status. We showed that loggerhead sea turtles represent potential umbrella species when coastal habitat frequented during breeding and non-breeding periods are used. However, it is equally important to protect areas supporting key resources of individual threatened species/taxa required for survival. In conclusion, our study presented a holistic approach to monitoring wildlife to better inform current and future conservation efforts, facilitating a shift towards more effective systematic conservation planning in protected area networks.

**Supplementary Materials:** The following supporting information can be downloaded at: https://www.mdpi.com/article/10.3390/drones6100291/s1, Figure S1. Study region showing the detection of loggerhead sea turtles during the breeding season (green circles), non-breeding season (blue circles) and both seasons combined (red circles) based on (**a**) aerial drone surveys and (**b**) tracking datasets. Figure S2. Map of the study region showing turtle density groupings (1–10, 11–50, 51–100, >100 turtles/km) for the breeding and non-breeding periods from (**a,b**) aerial drone surveys and literature sources combined, (**c,d**) Aerial drone surveys only, and (**e,f**) literature sources only. Natura 2000 protected areas: dashed green polygons. Circles represent 2 km sections (cells) along coastline. Key (Density; top right in panel a) is the same across all panels. Figure S3. Distribution of density

groupings (1–10, 11–50, 51–100, >100 turtles/km) across sites for the breeding (light grey; $n_{1-10} = 115$, $n_{11-50} = 28$, $n_{51-100} = 8$, $n_{>100} = 11$) and non-breeding (black; $n_{1-10} = 152$, $n_{11-50} = 13$, $n_{51-100} = 1$, $n_{>100} = 1$) periods from aerial drone surveys and literature sources combined. Figure S4. Distribution of cells inside (light grey) and outside (black) MPAs. Cell counts were higher counts for sites in MPAs ($\chi^2 = 9.08$, df = 3, $p < 0.05$) for sites where turtles were detected in breeding, non-breeding, or overlapping periods. Detections are from aerial drone surveys and literature sources combined. Of cells containing no turtles, 12 were surveyed by aerial drones and 518 were not, with no significant difference being found in analyses excluding these cells; thus data presented here include all cells (with non-surveyed cells excluded: $\chi^2 = 9.24$, df = 3, $p < 0.05$). Figure S5. Barplots showing the number of cells that fall inside (light grey) and outside (black) of protected areas across densities during (**a**) breeding season, and (**c**) nonbreeding season. Barplots showing the number of cells of each habitat type (V = vegetated; S = submerged sandbanks; R = rocky outcrops and reefs; M = mud/silt) where turtles were detected during (**b**) breeding season, and (**d**) nonbreeding season. (**e**,**f**) Barplots showing (**e**) cells across densities and (**f**) cells of different habitat types among foraging turtles in the Gulf of Argostoli. All Argostoli cells were lacking protection. Natura 2000 protected areas: dashed green polygons. Circles represent 2 km sections along coastline. Figure S6. Argostoli Gulf (Kefalonia) showing turtle density groupings (1–10, 11–50, 51–100, >100 turtles/km) during (**a**) summer, and (**b**) autumn. Turtles in the gulf are foraging year-round showing turtle density groupings (1–10, 11–50, 51–100, >100 turtles/km) during (**a**) summer, and (**b**) autumn. Turtles in the gulf are foraging year-round. Figure S7. (**a**) Distribution of cells with one or more species detected across the study region. For distribution of individual groups see Supplementary Figures S6 and S7. (**b**,**c**) Number of cells (2 km sections) containing (**b**) cells containing zero to 6 species, that fall inside (light grey) and outside (black) protected areas. (**c**) Habitat type (white = vegetated, light grey = submerged sandbanks, dark grey = rocky outcrops and reefs, black = mud/silt) of cells containing multiple species. Species were more likely to fall within MPAs ($\chi^2 = 18.2$, df = 2, $p < 0.001$). Figure S8. Study region showing detected presence from aerial drone surveys and literature sources combined for (**a**) cetaceans, (**b**) sea turtles (green and loggerheads combined), (**c**) elasmobranchs, and (**d**) pinnipeds. Natura 2000 protected areas: dashed green polygons. Circles represent 2 km sections along coastline. Figure S9. Study region showing detected presence from aerial drone surveys only for (**a**) cetaceans, (**b**) sea turtles (green and loggerheads combined), (**c**) elasmobranch, and (**d**) pinnipeds. Natura 2000 protected areas: dashed green polygons. Circles represent 2 km sections along coastline. Figure S10. Total number of unprotected (black) cells containing one or more taxa (**a**) outside of PA status, and the number of additional cells that are protected (light grey) when extending PA coverage based on (**b**) all cells containing breeding turtles; (**c**) all cells containing non-breeding turtles; (**d**) cells containing records of either breeding or non-breeding turtles (combined). Figure S11. Number of kilometres of coastline inside or outside MPAs capturing taxa based on five scenarios: (1) all cells with one or more taxa (baseline); (2) all cells containing breeding turtles; (3) all cells containing non-breeding turtles; (4) all cells containing combined breeding and non-breeding turtles; (5) just overlapping cells containing breeding/non-breeding turtles (combined). Black: 3 taxa cells, dark grey: 2 taxa cells, light grey: 1 taxon cells. Figure S12. Total number of unprotected (black) cells containing one or more species (**a**) outside of PA status, and the number of additional cells that are protected (light grey) when extending PA coverage based on (**b**) all cells containing breeding turtles; (**c**) all cells containing non-breeding turtles; (**d**) cells containing records of either breeding or non-breeding turtles (combined). Figure S13. (**a**) Number of kilometres of coastline inside MPAs capturing species based on five scenarios: (1) all cells with one or more species (baseline); (2) all cells containing breeding turtles; (3) all cells containing non-breeding turtles; (4) all cells containing combined breeding and non-breeding turtles; (5) just overlapping cells containing breeding/non-breeding turtles (combined). Black: 6 species cells, light grey: 1 species cells, with intervals of 1 species between. (**b**) Number of additional kilometres required to extend protection effort outside MPAs based on the same four scenarios. (**c**) Number of additional kilometres required to extend protection effort regardless of MPA status based on the same four scenarios. Table S1. Assimilated records of marine megafauna in the survey region from literature and online sources. Table S2. Chi-square Goodness of Fit Test statistics for different categories of cell classes and the primary explanatory variable. Table S3. Chi-square Test statistics for the relationship between number of cells with multiple taxa encompassed under four scenarios: (1) breeding, (2) non-breeding, (3) combined breeding/non-breeding, and (4) overlap breeding/non-breeding, and whether these cells are inside MPAs, outside MPAs, or both.

**Author Contributions:** Conceptualization, G.S.; field work, L.C.D.D., S.R.B.N., K.A.K. and G.S.; formal analysis, L.C.D.D., S.R.B.N. and G.S.; writing—original draft preparation, L.C.D.D., S.R.B.N. and G.S.; writing—review and editing, L.C.D.D., S.R.B.N., G.S., K.A.K. and C.E.; supervision, G.S.; funding acquisition, G.S. and S.R.B.N. All authors have read and agreed to the published version of the manuscript.

**Funding:** This research was funded by Queen Mary University of London (UK) and the London Natural Environment Research Council (NERC) Doctoral Training Partnership (DTP) awarded to S.R.B.N.

**Institutional Review Board Statement:** Not applicable.

**Informed Consent Statement:** Not applicable.

**Data Availability Statement:** Data will be made available in Mendeley Data or Open Science Framework on acceptance for publication.

**Acknowledgments:** Permission for fieldwork was given by the Greek Ministry of Environment (Permit: 151503/162 and 181806/941). We also thank Kostas Poirazidis, Joshua Simcock, Kesten Laverty, and Liam Nash for providing fieldwork assistance.

**Conflicts of Interest:** The authors declare no conflict of interest. The funders had no role in the design of the study; in the collection, analyses, or interpretation of data; in the writing of the manuscript, or in the decision to publish the results.

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
