# Peer review of "Aerial Drone Surveys Reveal the Efficacy of a Protected Area Network for Marine Megafauna and the Value of Sea Turtles as Umbrella Species"

_drones, doi:10.3390/drones6100291_

Round 1
Reviewer 1 Report
General comment
A solid manuscript that needs minor amendments throughout, and some slight rethinking in parts of the discussion. A lot of work went into this manuscript, and it paid off. All of the comments below are easy to fix, but just need a bit of thought.
Specific comments
Line 20: “with aerial drones consistently recording ~5 times more turtles”…5x more than which other method? Comparator is not clear.
Line 20 “demonstrating the value of combing approaches”…needs a context. Combined approaches are valuable for many things so just add a few words at the end.
Line 54: However, all of these approaches have inherent biases in location….” I’d soften this a little to not antagonise anyone. Something like, ‘Despite the invaluable insights garnered through such studies, all of these approaches have inherent biases in….”. That avoids the oversimplified ‘your approach rubbish, my approach excellent’ accidental pitfall. Also, biases and bias in the same sentence feels a little clunky so needs a minor tweak.
Line 92: “We hypothesised that aerial drone surveys would provide more balanced coverage of marine megafauna distributions across the network (i.e. inside vs outside PAs) than remote tracking of individual animals, using sea turtles as a case study.” Makes sense but is slightly site specific. For example if you had a huge oceanic MPA drones would compare unfavourably with tracking. It also depends of how many drone surveys you can do in a given period and the number of animals fitted with transmitters. So, I’m not sure you can set it up as a one-size fits all hypotheses that drones will perform more favourably than tracking. Perhaps I am not reading the phrase ‘more balanced’ in the way it is intended. No major issues here, just a bit of rewording needed. To me it is showing what extra benefit drones bring to the table on top of tracking, rather than setting it up as which method is better as there is no singular answer to that question.
Line 99: Old school recommendation but I’d change last lines to past tense. So “Our results are expected to demonstrate the value of….” Could become “We expected our results to demonstrate’ as the study is in the past. Pedantic I know.
Regarding study sites (and elsewhere), it would be nice to make reference to the original study of loggerhead foraging around Argostoli, as it has relevance overall to this paper:
Houghton, Jonathan D.R., Woolmer, Andrew and Hays, Graeme C. 2000, Sea turtle diving and foraging behaviour around the Greek Island of Kefalonia, Journal of the Marine Biological Association of the United Kingdom, vol. 80, no. 4, pp. 761-762, doi: 10.1017/S002531540000271X.
Line 196: “we examined how the coverage of the MPA network could be enhanced”…this could mean spatial or temporal coverage, or both. Needs clarifying a little.
Line 199: “We also explored the effectiveness of using sea turtles as an umbrella species for multiple taxa in the breeding and non-breeding seasons, both inside and outside of MPAs”. Sounds nice, but a little bit more detail needed. Does this reflect some form of analysis or is it simply a point of discussion later in the manuscript. Essentially, you need a line or two to explain what this means in the context of the methods.
Line 245: “Furthermore, relative numbers (low to high) of turtles were positively correlated (r = 0.48, t = 4.3, df = 62, P < 0.0001), but with approximately 5 times more turtles being consistently recorded in drone surveys compared to tracking. Thus, we provide a means of using tracking datasets to indicate minimum potential turtle numbers”. Apologies but I am not following how this finding is generalisable. This depends on how many turtles have been tracked surely or am I missing something? In my head, drones supplement tracking efforts here by providing information of how representative tracking data are of distribution and habitat utilisation is a given area. But I can’t see how tracking data provide information on potential turtle numbers (i.e. the number of individuals recorded with drones will not always be five times greater than the number of tracked animals at all sites).
Line 331: “demonstrating that sea turtles are highly viable umbrella species when based on combined breeding and non-breeding areas”. Just needs a slight tweak in wording as breeding and non-breeding areas combined (in a literal sense) is everywhere in the ocean as all sites fall into one of these categories.
Line 347: “…to remove potential bias associated with individual approaches alone”. Yes, but drones themselves have potential biases. As it stands, this sentence essentially reads as drones can make any data set perfect (accidentally, I appreciate). This still leaves the reader with a ‘drones good, other methods a bit lame’ feeling. Just a slight tweak again needed.
Line 349: “existing network of Natura 2000 Pas”…This is not the only existing network of protected areas under Natura 2000, but this is accidentally how it reads.
Line 359: “Sea turtles forage both on reefs (for sponges) and mine for molluscs in shallow submerged sandbanks and muddy substrate [74, 75].” This is not the diet of all sea turtles which is how it is written. When referring to greens there is also seagrass etc.
Line 367: “Fine-scale mapping with field validation through snorkelling or remotely operated
underwater vehicles (ROV) could help refine our observations and identify potential gaps”. Yes, but this is a very localised recommendation. The journal will be thinking about the general reader and citations so make sure to emphasise how your approach is transferable. Otherwise, it becomes a description of just this site, which lessens its appeal as a methods paper.
Line 394: “This study showed that loggerhead sea turtles represent viable umbrella species for 394
other marine megafauna, particularly when both breeding and non-breeding areas are 395
selected for protection.” That is pushing the discussion a bit far. Essentially, you are making the case that all sea turtles (from leatherbacks to flatbacks) in all areas are excellent umbrella species for marine megafauna globally. They worked well here and you should encourage others to explore this approach, but the sentence you have at present goes beyond your results.
Line 396: “This is logical because breeding areas are primarily restricted to certain habitats with characteristics generally not associated with foraging, and so might not support the resources required by other taxa as effectively”. Not sure I follow this. Does this mean, because there may not be prey for a benthic omnivore that it is logical that prey for pelagic sharks or cetaceans would also not be present? That doesn’t necessarily follow so becomes a bit of a guess unfortunately, rather than a logical assertion. Be careful to discuss what the data show, rather than what you want the data to show (we are all guilty of this).
Line 398: “This is logical because breeding areas are primarily restricted to certain habitats with characteristics generally not associated with foraging, and so might not support the resources required by other taxa as effectively”. The different faunal groups you have worked on are tied into different foodwebs to some extend (pelagically derived and benthically derived). As you don’t really go into diet at all in this study I would stay away from statement about trophic levels. Foraging is not the same as feeding so make sure that word is used tightly as well. Just feels like you are stretching a bit far in the final stages of the discussion.
Line 401: “As the seven sea turtle species forage on different items”. Foraging is the process of searching for food rather than ingesting it. So, animals forage for prey, and feed on it. Pedantic, but an important distinction.
Line 409: Change ‘;’ to ‘.’
Line 416: “….and the need for active monitoring inside and outside MPAs to facilitate informed science-based management”. Accidentally, this sentence infers that studies that are based on monitoring within MPAs alone are not ‘science-based management’. So, you are inadvertently claiming that this paper has invented science-based management. See what I mean? Slight rewording needed.
Line 423: “We confirmed that loggerhead sea turtles are viable umbrella species, as long as both breeding and non-breeding areas are used”. Non-breeding areas needs refining a bit as that (although you don’t mean it like this) is everywhere else in the ocean that this population doesn’t breed in. Needs a little think how to get around that potential trap.
Line 424: Change ‘;’ to ‘.’
Reviewer 2 Report
I feel this study contributes to the much needed re-evaluation of traditional methods of conservation management in light of climate change and more effective ecosystem-based approaches. All while demonstrating the unique capacity of novel drone technology to enhance management practices. Aside from some minor clarifications for the reader and grammatical corrections, this study is a delight to welcome to the scientific literature in this genre. Great job!

Reviewer 3 Report
The manuscript brings an interesting application of marine megafauna records by drone. Overall the article is well written and presented. I recommend publishing, after some major revisions:
1 - The authors found a relationship between the number of records of sea turtles with elasmobranchs, cetaceans and pinnipeds. However, this relationship is not enough to state that sea turtles serve as an umbrella or surrogate species for the conservation of other taxa, considering the different patterns of aggregation involved. Authors should better present the limits/bias of their claims. They should emphasize in the abstract, results and conclusions that the results are valid only for coastal areas.
2 - Maps need a bathymetric reference (at least one isobath). Remove the border lines from the colored circles, they affect the results visualization. Ideally, keep the cell format colored and borderless. Also dashed green polygons could just be green polygons.
3 - Was the sampling effort homogeneous? If yes, explain further. If not, include the effort distribution (e.g. number or hours of overflight in each cell) and explain the relativity in the results. I understand that the sample drawing is referenced, however, I wouldn't want to read another paper for some background information.
I would like to congratulate the authors for contributing to the disruption of sampling methods by applying drone technology to support marine spatial planning actions.
Thank you for the opportunity, I remain at your disposal for future contributions.
Reviewer 4 Report
I find this work very interesting and indicative of the use of drones as a very good tool for monitoring large fauna species in protected areas.
The paper is a bit too long and contains too many figures. Perhaps it would be possible to reduce this? This is my the most important recommendetion.
On the other hand, the paper has a lot of literature items and generally refers to the current state of knowledge, but despite 117 position it is surprising that the following item is missing, which fits perfectly thematically:
Drone Technology for Monitoring Protected Areas in Remote and Fragile Environments.
B Bollard, A Doshi, N Gilbert, C Poirot, L Gillman
Drones 6 (2), 42 https://doi.org/10.3390/drones6020042
In addition, some statements are perhaps not very well supported by the literature, e.g., in lines: 48-50 Authors wrote: “While some species aggregate in large numbers at breeding 48 or foraging rounds in predictable locations [e.g., pinnipeds, 20, sea turtles, 21], other species are mostly pelagic, and are often solitary [e.g., whales, 22, sharks, 23].”
In the case of pinnipeds, a much more fitting paper is:
Breeding colony dynamics of southern elephant seals at Patelnia Point, King George Island, Antarctica
K Fudala, RJ Bialik
Remote Sensing 12 (18), 2964
whose research on southern elephant seals was carried out at Antarctic Specially Protected Areas no. 128 using drones DJI Inspire 2.
Such cases can be found in this work, and one would have to wonder whether 117 items of literature is the right amount, or whether it would be better to reduce it but still be more to the point and up to date.
Round 2
Reviewer 3 Report
Dear Authors,
After the review the work is much better and is ready to be published.
Well done!
Thank you!